METHODS AND RESOURCES

# Multiplex qPCR discriminates variants of concern to enhance global surveillance of SARS-CoV-2

**Chantal B. F. Vogels**[1☺*], **Mallery I. Breban**[1☺], **Isabel M. Ott**[1], **Tara Alpert**[1], **Mary E. Petrone**[1], **Anne E. Watkins**[1], **Chaney C. Kalinich**[1], **Rebecca Earnest**[1], **Jessica E. Rothman**[1], **Jaqueline Goes de Jesus**[2], **Ingra Morales Claro**, **Giulia Magalhães Ferreira**[2,3], **Myuki A. E. Crispim**[4], **Brazil-UK CADDE Genomic Network**[¶], **Lavanya Singh**[5], **Houriiyah Tegally**[5], **Ugochukwu J. Anyaneji**[5], **Network for Genomic Surveillance in South Africa**[¶], **Emma B. Hodcroft**[6], **Christopher E. Mason**[7], **Gaurav Khullar**[7], **Jessica Metti**[7], **Joel T. Dudley**[7], **Matthew J. MacKay**[7], **Megan Nash**[7], **Jianhui Wang**[8], **Chen Liu**[8], **Pei Hui**[8], **Steven Murphy**[9], **Caleb Neal**[9], **Eva Laszlo**[9], **Marie L. Landry**[10], **Anthony Muyombwe**[11], **Randy Downing**[11], **Jafar Razeq**[11], **Tulio de Oliveira**[5], **Nuno R. Faria**[2,12,13], **Ester C. Sabino**[2], **Richard A. Neher**[14,15], **Joseph R. Fauver**[1*‡], **Nathan D. Grubaugh**[1,16*‡]

**1** Department of Epidemiology of Microbial Diseases, Yale School of Public Health, New Haven, Connecticut, United States of America, **2** Departamento de Molestias Infecciosas e Parasitarias and Instituto de Medicina Tropical da Faculdade de Medicina da Universidade de São Paulo, São Paulo, Brazil, **3** Laboratório de Virologia, Instituto de Ciências Biomédicas, Universidade Federal de Uberlândia, Uberlândia, Minas Gerais, Brazil, **4** Fundação Hospitalar de Hematologia e Hemoterapia do Amazonas, Manaus, Brazil, **5** KwaZulu-Natal Research Innovation and Sequencing Platform (KRISP), School of Laboratory Medicine & Medical Sciences, University of KwaZulu-Natal, Durban, South Africa, **6** Institute of Social and Preventive Medicine, University of Bern, Bern, Switzerland, **7** Tempus Labs, Chicago, Illinois, United States of America, **8** Department of Pathology, Yale University School of Medicine, New Haven, Connecticut, United States of America, **9** Murphy Medical Associates, Greenwich, Connecticut, United States of America, **10** Departments of Laboratory Medicine and Medicine, Yale School of Medicine, New Haven, Connecticut, United States of America, **11** Connecticut State Department of Public Health, Rocky Hill, Connecticut, United States of America, **12** MRC Centre for Global Infectious Disease Analysis, J-IDEA, Imperial College London, London, United Kingdom, **13** Department of Zoology, University of Oxford, Oxford, United Kingdom, **14** Biozentrum, University of Basel, Basel, Switzerland, **15** Swiss Institute of Bioinformatics, Lausanne, Switzerland, **16** Department of Ecology and Evolutionary Biology, Yale University, New Haven, Connecticut, United States of America

☺ These authors contributed equally to this work.
‡ These authors are joint senior authors on this work.
¶ Membership of the Brazil-UK CADDE Genomic Network and the Network for Genomic Surveillance in South Africa (NGS-SA) is provided in the Acknowledgments.
* chantal.vogels@yale.edu (CBFV); joseph.fauver@yale.edu (JRF); nathan.grubaugh@yale.edu (NDG)

**Data Availability Statement:** Genomic data are available on GISAID (see S2 Data for accession numbers). All RT-qPCR data are included in this article and the supporting information.

## Abstract

With the emergence of Severe Acute Respiratory Syndrome Coronavirus 2 (SARS-CoV-2) variants that may increase transmissibility and/or cause escape from immune responses, there is an urgent need for the targeted surveillance of circulating lineages. It was found that the B.1.1.7 (also 501Y.V1) variant, first detected in the United Kingdom, could be serendipitously detected by the Thermo Fisher TaqPath COVID-19 PCR assay because a key deletion in these viruses, spike Δ69–70, would cause a "spike gene target failure" (SGTF) result. However, a SGTF result is not definitive for B.1.1.7, and this assay cannot detect other variants of concern (VOC) that lack spike Δ69–70, such as B.1.351 (also 501Y.V2), detected in

**Funding:** This work was funded by CTSA Grant Number TL1 TR001864 (TA and MEP), Wellcome Trust and Royal Society Sir Henry Dale Fellowship (204311/Z/16/Z; NRF), a Medical Research Council-São Paulo Research Foundation CADDE partnership award (MR/S0195/1 and FAPESP 18/14389-0; NRF), Fast Grant from Emergent Ventures at the Mercatus Center at George Mason University (NDG), and CDC Contract # 75D30120C09570 (NDG). The funders had no role in study design, data collection and analysis, decision to publish, or preparation of the manuscript.

**Competing interests:** The authors have declared that no competing interests exist.

**Abbreviations:** cDNA, complementary DNA; COVID-19, Coronavirus Disease 2019; Ct, cycle threshold; IRB, Institutional Review Board; RT-qPCR, reverse transcription quantitative PCR; SARS-CoV-2, Severe Acute Respiratory Syndrome Coronavirus 2; SGTF, spike gene target failure; VOC, variants of concern; VOI, variants of interest.

South Africa, and P.1 (also 501Y.V3), recently detected in Brazil. We identified a deletion in the ORF1a gene (ORF1a Δ3675–3677) in all 3 variants, which has not yet been widely detected in other SARS-CoV-2 lineages. Using ORF1a Δ3675–3677 as the primary target and spike Δ69–70 to differentiate, we designed and validated an open-source PCR assay to detect SARS-CoV-2 VOC. Our assay can be rapidly deployed in laboratories around the world to enhance surveillance for the local emergence and spread of B.1.1.7, B.1.351, and P.1.

## Introduction

Broadly accessible and inexpensive surveillance methods are needed to track Severe Acute Respiratory Syndrome Coronavirus 2 (SARS-CoV-2) variants of concern (VOC) around the world. While sequencing is the gold standard to identify circulating SARS-CoV-2 variants, routine genomic surveillance is not available in many locations primarily due to a lack of resources and expertise. In the current situation, with the identification of the VOC B.1.1.7, B.1.351, and P.1 [1–3] and with the likelihood that more will emerge, a lack of genomic surveillance leaves public health authorities with a patchy and skewed picture to inform decision-making. The discovery of B.1.1.7 variants causing spike gene target failure (SGTF) results when tested using the TaqPath PCR assay provided labs in the United Kingdom and throughout Europe with a ready-made, simple tool for tracking the frequencies of this variant [4–7]. As B.1.1.7 spread to other countries, TaqPath SGTF results were used as a frontline screening tool for sequencing and an approximation for B.1.1.7 population frequency [8]. These findings highlight the usefulness of a PCR assay that produces distinctive results when targeting variants in virus genomes for both tracking and sequencing prioritization.

The TaqPath assay was not specifically designed for SARS-CoV-2 variant surveillance, and it has several limitations. The 6-nucleotide deletion in the spike gene at amino acid positions 69 and 70 (spike Δ69–70) that causes the TaqPath SGTF is also present in other SARS-CoV-2 lineages (**Fig 1**, **S1 Table**), most notably in Pango lineages B.1.258 detected throughout Europe and in B.1.375 detected primarily in the United States [9,10], meaning that SGTF results are not definitive for B.1.1.7. Furthermore, too much focus on TaqPath SGTF results will leave blind spots for other emerging SARS-CoV-2 VOC that do not have spike Δ69–70. In particular, B.1.351 and P.1, which were recently discovered in South Africa and Brazil [11,12], respectively, may also be more transmissible and contain mutations that could help to evade immune responses [11–13]. For all of these reasons, a PCR assay specifically designed for variant surveillance would help to fill in many of the gaps about their distribution and frequency.

## Results

We analyzed over 400,000 SARS-CoV-2 genomes on GISAID and used custom Nextstrain builds [14] to identify that a 9-nucleotide deletion in the ORF1a gene at amino acid positions 3675–3677 (ORF1a Δ3675–3677) occurs in the B.1.1.7, B.1.351, and P.1 variants, but is only found in 0.03% (103/377,011) of all other genomes (**Fig 1**, **S1 Table**). Within the B.1.351 lineage, however, 18.4% of the sequences do not have ORF1a Δ3675–3677 (**S1 Table**). By designing a PCR assay that targets both ORF1a Δ3675–3677 and spike Δ69–70 (**Fig 1A**), we can detect most viruses from all 3 current VOC (ORF1a results, **Fig 1B–1E**), differentiate B.1.1.7 (ORF1a and spike results, **Fig 1D–1F**), and provide results similar to TaqPath SGTF to compare datasets (spike results).

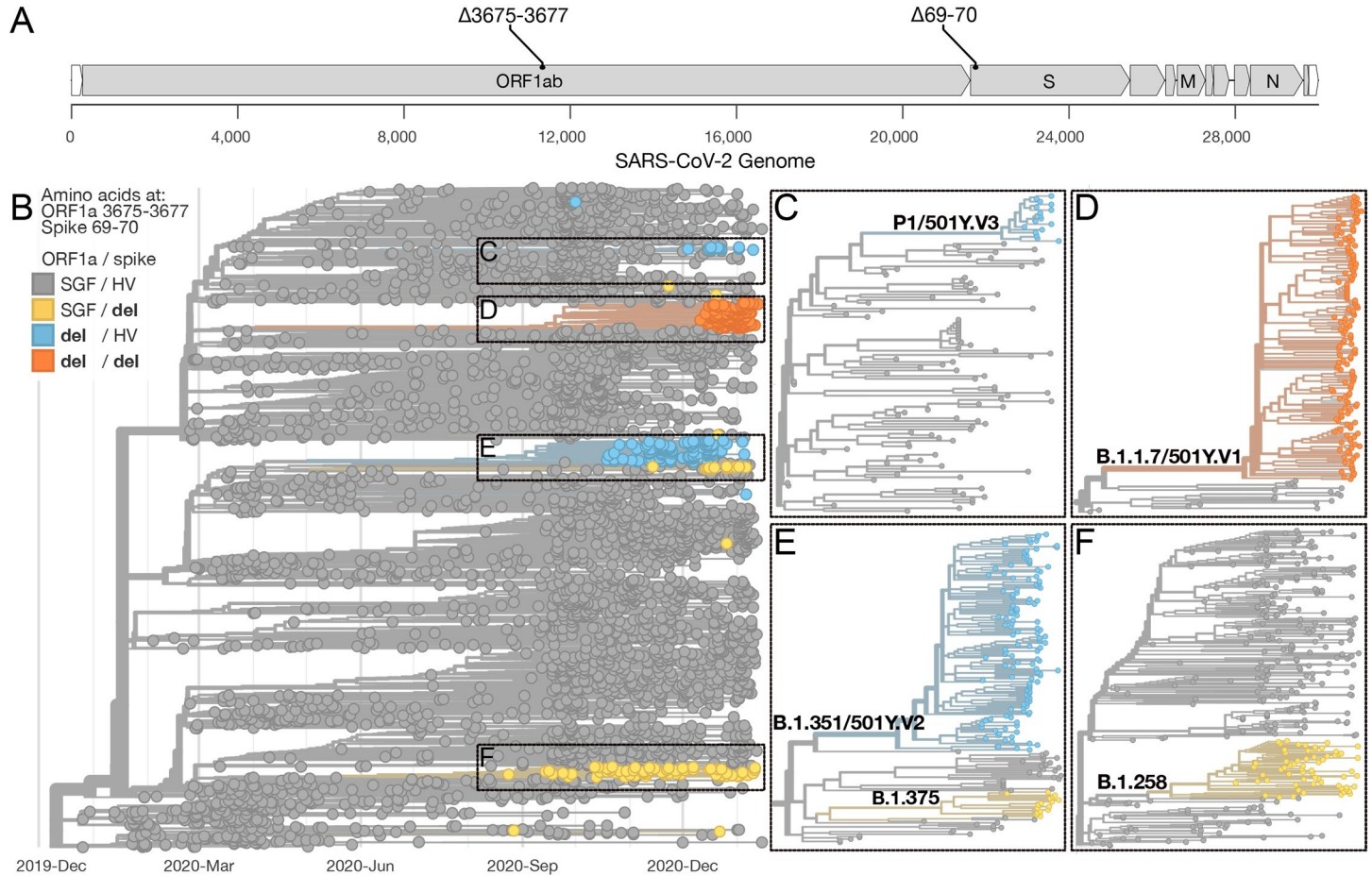

**Fig 1. Identification of genome targets to differentiate between B.1.1.7, B.1.351, P.1, and other SARS-CoV-2 lineages.** (A) Location on the SARS-CoV-2 genome where the targeted deletions in the ORF1a gene at amino acid positions 3675–3677 (Δ3675–3677) and the spike gene at amino acid positions 69–70 (Δ69–70) occur. (B) The Nextstrain "global build" (nextstrain.org/ncov/global) accessed on January 22, 2021 showing the phylogenetic representation of 4,046 SARS-CoV-2 genomes colored by the presence of deletions at amino acid positions ORF1a 3575–3677 and spike 69–70. (C–F) Zooms of large SARS-CoV-2 clades, which include the VOC B.1.1.7, B.1.351, and P.1, containing 1 or both deletions. A list of SARS-CoV-2 genomes used in the analysis is available in **S1 Data**. SARS-CoV-2, Severe Acute Respiratory Syndrome Coronavirus 2; SGTF, spike gene target failure; VOC, variants of concern.

To create a multiplexed reverse transcription quantitative PCR (RT-qPCR) screening assay for the B.1.1.7, B.1.351, and P.1 variants, we designed 2 sets of primers that flank each of ORF1a Δ3675–3677 and spike Δ69–70 and probes specific to the undeleted "wild-type" sequences. As a control, we included the CDC N1 (nucleocapsid) primer and probe set that will detect both the wild-type and variant viruses [15]. As designed, testing SARS-CoV-2 RNA that contains ORF1a Δ3675–3677 and/or spike Δ69–70 will generate undetected cycle threshold (Ct) values with the specific PCR target sets as the probes cannot anneal to the deleted sequences, but will have "positive" N1 Ct values. This configuration ensures that target failures are likely due to the presence of deletions and that there is sufficient virus RNA for sequencing confirmation. Our RT-qPCR conditions are highly similar to our previously published SARS-CoV-2 multiplex assay [16], and a detailed protocol is openly available [17].

We evaluated the analytical sensitivity of our multiplexed RT-qPCR assay using synthetic RNA designed based on the original Wuhan-Hu-1 sequence and a B.1.1.7 sequence (England/ 205041766/2020). As the B.1.1.7 sequence contains both ORF1a Δ3675–3677 and spike Δ69– 70 and the Wuhan-Hu-1 sequence contains neither deletion, using these RNAs allows us to

**Table 1. Analytical sensitivity of the multiplexed RT-qPCR assay to screen for VOC using primer/probe sets targeting key deletions.**

| RNA | Concentration | N1-FAM | | | ORF1a-Cy5 | | | Spike-HEX | | |
|-----|---------------|--------|------|------|-----------|------|------|-----------|------|------|
| Wuhan-Hu-1 | 100 copies/μL | 30.4 | 30.5 | 30.4 | 31.1 | 31.2 | 31.1 | 30.8 | 30.8 | 30.9 |
| | 50 copies/μL | 31.5 | 31.5 | 31.4 | 32.1 | 32.1 | 32.0 | 31.9 | 31.6 | 31.7 |
| | 25 copies/μL | 32.3 | 32.7 | 32.7 | 33.0 | 33.1 | 33.2 | 32.8 | 32.8 | 33.0 |
| | 12 copies/μL | 33.3 | 33.7 | 33.3 | 34.1 | 34.2 | 34.2 | 34.0 | 33.6 | 34.1 |
| | 6 copies/μL | 34.5 | 34.3 | 34.9 | 35.1 | 35.2 | 35.4 | 35.1 | 34.3 | 35.6 |
| | 3 copies/μL | 35.0 | 37.0 | 36.1 | 35.6 | 36.6 | 36.8 | 35.4 | 36.1 | 35.8 |
| | 1 copy/μL | 37.0 | 37.0 | 35.7 | 37.2 | 36.7 | 37.0 | 36.4 | 36.4 | 36.3 |
| B.1.1.7 | 100 copies/μL | 29.1 | 29.2 | 29.2 | ND | ND | ND | ND | ND | ND |
| | 50 copies/μL | 30.1 | 30.2 | 30.2 | ND | ND | ND | ND | ND | ND |
| | 25 copies/μL | 31.1 | 31.2 | 31.2 | ND | ND | ND | ND | ND | ND |
| | 12 copies/μL | 32.5 | 32.2 | 32.1 | ND | ND | ND | ND | ND | ND |
| | 6 copies/μL | 33.0 | 33.2 | 33.1 | ND | ND | ND | ND | ND | ND |
| | 3 copies/μL | 33.7 | 34.6 | 34.0 | ND | ND | ND | ND | ND | ND |
| | 1 copy/μL | 35.0 | 35.2 | 35.3 | ND | ND | ND | ND | ND | ND |

Listed are Ct values for the 3 primer–probe sets targeting the SARS-CoV-2 nucleocapsid (N1-FAM), ORF1a 3675–3766 deletion (ORF1a-Cy5), and spike 69–70 deletion (Spike-HEX). Two-fold dilutions of synthetic control RNA (Wuhan-Hu-1 and B.1.1.7) were tested in triplicate.

Wuhan-Hu-1 = Twist synthetic RNA control 2; B.1.1.7 = Twist synthetic RNA control 14; N1-FAM = CDC N1 primer–probe set targeting the nucleocapsid with FAM fluorophore [15]; ORF1a-Cy5 = Yale primer–probe set targeting the ORF1a gene 3675–3677 deletion with Cy5 fluorophore [17]; Spike-HEX = Yale primer–probe set targeting the spike gene 69–70 deletion with HEX fluorophore [17].

CT, cycle threshold; RT-qPCR, reverse transcription quantitative PCR; VOC, variants of concern.

fully evaluate the designed primer and probe sets. We tested a 2-fold dilution series from 100 copies/μL to 1 copy/μL for both RNA controls in triplicate (**Table 1**). Using the Wuhan-Hu-1 RNA, we found similar detection (within 1 Ct) across all 3 N1, ORF1a, and spike targets, and all 3 could detect virus RNA at our lowest concentration of 1 copy/μL, indicating that our primer and probes sets were efficiently designed. Using the B.1.1.7 RNA, we again could detect the RNA down to 1 copy/μL with the N1 set, but did not detect any concentration of the virus RNA with the ORF1a and spike sets, confirming the expected "target failure" signature when testing viruses containing both ORF1a Δ3675–3677 and spike Δ69–70. Overall, our PCR screening assay could easily differentiate between SARS-CoV-2 RNA with and without the ORF1a and spike deletions by comparing the Ct values to the N1 control.

Next, we validated our multiplex RT-qPCR variant screening assay using 1,361 known Coronavirus Disease 2019 (COVID-19) clinical samples that we have previously sequenced in our laboratories in the US, Brazil, and South Africa (**Fig 2**, **S2 Data**). We tested 481 samples from SARS-CoV-2 lineages without either ORF1a Δ3675–3677 and spike Δ69–70 (classified as "other" lineage), 606 samples with both ORF1a Δ3675–3677 and spike Δ69–70 deletions, 227 samples with only the ORF1a Δ3675–3677 deletion, and 47 samples with only the spike Δ69–70 deletion. Of the samples without both deletions (expected outcome = detection with all 3 primer/probe sets), 1.2% (6/481) were P.2 (variants of interest [VOI]), 2.1% (10/481) were B.1.427 (VOC), 4.2% (20/481) were B.1.429 (VOC), and 92.5% (445/481) were other lineages not of current interest (**Fig 2A**). Of the samples with both ORF1a Δ3675–3677 and spike Δ69–70 deletions (expected outcome = target failure with both the ORF1a and spike sets), 98.2% (595/606) were B.1.1.7 (VOC), 1.3% (8/606) were B.1.525 (VOI), and 0.5% (3/606) were other lineages not of current interest (**Fig 2B**). Of the samples with ORF1a Δ3675–3677, 11.5% (26/227) were B.1.351 (VOC), 9.3% (21/227) were P.1 (VOC), 70.9% (161/227) were B.1.526 (VOI; includes B.1.526.1 and B.1.526.2), and 8.4% (19/227) were other lineages not of current interest

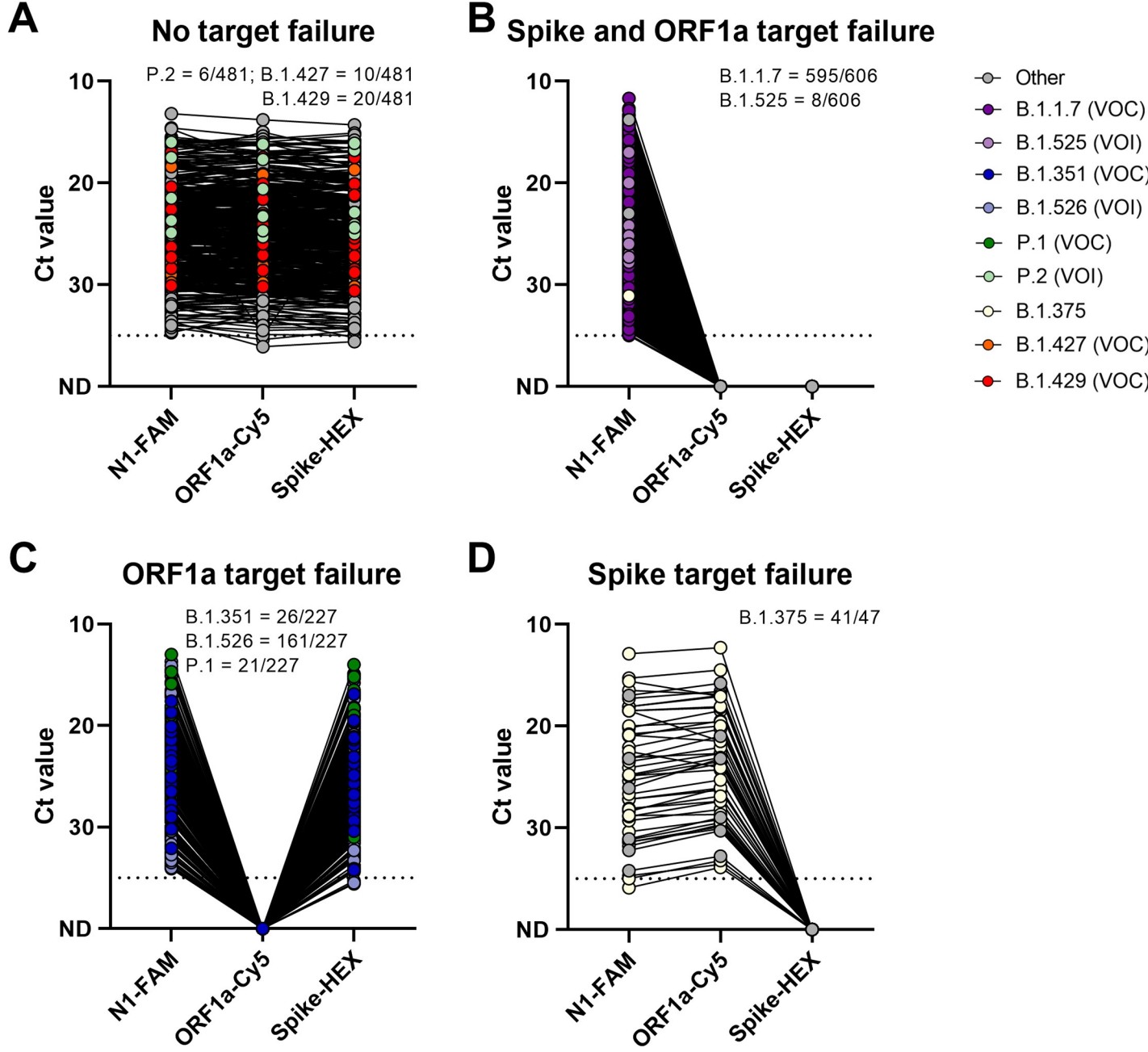

**Fig 2. ORF1a and spike target failure used to differentiate between SARS-CoV-2 VOC B.1.1.7, B.1.351, P.1, and other lineages currently not of concern.** (**A**) Lineages (including B.1.427 and B.1.429) without target failure are detected by all 3 targets of the multiplex RT-qPCR assay. (**B**) Double target failure indicates the presence of the ORF1a Δ3675–3677 and spike Δ69–70 deletions and can be used to identify potential B.1.1.7 variants. (**C**) ORF1a target failure indicates the presence of the ORF1a Δ3675–3677 deletion, present in B.1.351 and P.1 VOC. (**D**) Spike target failure indicates presence of the spike Δ69–70 deletion, which is present in various lineages, including B.1.375. Shown are the Ct values for the N1, ORF1a, and spike primer–probe sets, with lines connecting Ct values obtained with the 3 sets for the same specimen. The dotted line indicates the limit of detection. Data used to make this figure can be found in **S2 Data**. Ct, cycle threshold; RT-qPCR, reverse transcription quantitative PCR; SARS-CoV-2, Severe Acute Respiratory Syndrome Coronavirus 2; VOC, variants of concern; VOI, variants of interest.

(**Fig 2C**). Of the samples with only the spike Δ69–70 deletion, 87.2% (41/47) were B.1.375, and 12.8% (6/47) were other lineages not of current interest (**Fig 2D**). Importantly, unlike the Taq-Path assay SGTF results, we could differentiate between B.1.1.7 and other variants that only have the spike deletion, such as B.1.375 that is not currently a VOC/VOI.

When splitting up our findings by country, we tested a total of 1,290 samples from the US, 23 from Brazil, and 48 from South Africa. Within the US, the majority of samples were from Connecticut, Illinois, New Jersey, and New York. Of the samples without target failure, 2.2% (10/451) were B.1.427 (VOC), 4.4% (20/451) were B.1.429 (VOC), and 93.3% (421/451) were other lineages not of current interest. Of the samples with double target failure, 98.2% (595/606) were B.1.1.7 (VOC), 1.3% (8/606) were B.1.525 (VOI), and 0.5% (3/606) were other lineages not of current interest. Of the samples with ORF1a target failure, 1.1% (2/186) were B.1.351 (VOC), 2.7% (5/186) were P.1 (VOC), 86.6% (161/186) were B.1.526 (VOI; includes B.1.526.1 and B.1.526.2), and 9.7% (18/186) were other lineages. Of the samples with spike target failure, 87.2% (41/47) were B.1.375, and 12.8% (6/47) were other lineages. Thus, if we used ORF1a target failure (with or without spike target failure) to identify high priority samples for sequencing, we would have confirmed 100% of the B.1.1.7, B.1.351, and P.1 samples in the US. In addition, we would have detected 2 new VOI (B.1.525 and B.1.526, both containing the E484K mutation) [18] while triaging 34.9% (451/1,290) of the samples.

In Brazil, we tested samples from the cities of São Paulo and Manaus. Of the samples without target failure, 85.7% (6/7) were P.2 (VOI), and 14.3% (1/7) were other lineages. Of the samples with ORF1a target failure, 100% (16/16) were P.1 (VOC). Within South Africa, samples were tested from the KwaZulu-Natal Province. Of the samples without target failure, 100% (23/23) samples belonged to other lineages. Of the samples with ORF1a target failure, 96% (24/25) were B.1.351 (VOC), and 4% (1/25) belonged to other lineages. Thus, our clinical results demonstrate how our multiplex RT-qPCR assay can detect potential SARS-CoV-2 VOC and/or VOI from a variety of settings to prioritize samples for sequencing.

## Discussion

The rapid emergence of the SARS-CoV-2 VOC necessitates an immediate roll out of surveillance tools. Although whole genome sequencing is required to definitively identify specific variants, resource and capacity constraints can limit the number of samples that can be sequenced. The Thermo Fisher TaqPath assay has demonstrated the value of PCR for variant surveillance, but it is limited to B.1.1.7 and cannot differentiate between other viruses containing spike Δ69–70. By targeting 2 different large nucleotide deletions, ORF1a Δ3675–3677 and spike Δ69–70, we demonstrate that our multiplex PCR can rapidly screen for B.1.1.7, B.1.351, and P.1 variants, detect other VOI, and differentiate between most non-VOC. Thus, our multiplex RT-qPCR variant screening assay can be used to prioritize samples for sequencing and as a surveillance tool to help monitor the distribution and population frequency of suspected variants.

There are some limitations to our study as presented here. First, we initially observed autofluorescence of the N1 primer–probe set when testing negative template controls. By lowering the N1 primers and probe concentrations to 200 nM and 100 nM per reaction, respectively, we have reduced autofluorescence to levels above our threshold of Ct 35 (autofluorescence detected in 3/48 reactions with average Ct 39.6). Importantly, this PCR assay should only be used to screen known SARS-CoV-2 positive clinical samples for the presence of key deletions found in VOC (where autofluorescence will not be a factor), and it should not be used as a primary clinical diagnostic. We also suggest using a N1 threshold Ct of 30 to 35 for calling target failures in the ORF1a and spike sets and performing whole genome sequencing to confirm the identity of variants. For instance, the threshold for the system used in Brazil was lowered to Ct 30 as there did not seem to be consistent detection of all 3 sets above that threshold. Thus, the threshold may differ between used RT-qPCR kits and instruments and needs to be determined by individual laboratories.

Second, our assay will not be 100% sensitive and/or specific to all VOC or VOI, especially given the rapid emergence of new mutations and variants. For instance, we detected other recently emerged virus lineages with double target failure (notably B.1.525) and ORF1a target failure (notably B.1.526), which are not currently VOC. B.1.525 and B.1.526, however, are recognized as VOI as some contain the E484K and other key spike mutations that require further evaluation of their implications for transmissibility and immune escape [18]. Moreover, B.1.427 and B.1.429 (no gene target failure) have been recognized as VOC since the initial development of our assay [19], but neither cause ORF1a or spike target failures. Lastly, there is a monophyletic clade within the B.1.351 lineage that has ORF1a Δ3675–3677 filled back in, perhaps due to recombination with viruses that did not have the deletion. Depending on changing scenarios, presence as well as absence of the ORF1a and spike deletions can both be used to target for potential VOC or VOI, or to estimate their frequencies over time. Thus, these examples demonstrate that continuous monitoring of the lineages that contain the ORF1a Δ3675–3677 and spike Δ69–70 deletions will be necessary to direct the local uses of our assay.

## Materials and methods

### Ethics

**United States.** The Institutional Review Board (IRB) from the Yale University Human Research Protection Program determined that the RT-qPCR testing and sequencing of de-identified remnant COVID-19 clinical samples conducted in this study is not research involving human patients (IRB Protocol ID: 2000028599).

**Brazil.** Sample collection and genetic characterization were approved under the Brazilian National IRB (CONEP) CAAE 30101720.1.0000.0068.

**South Africa.** We used de-identified remnant nasopharyngeal and oropharyngeal swab samples from patients testing positive for SARS-CoV-2 by RT-qPCR from public health and private medical diagnostics laboratories in South Africa. The project was approved by University of KwaZulu-Natal Biomedical Research Ethics Committee (protocol reference no. BREC/00001195/2020; project title: COVID-19 transmission and natural history in KwaZulu-Natal, South Africa: epidemiological investigation to guide prevention and clinical care). Individual participant consent was not required for the genomic surveillance. This requirement was waived by the research ethics committees.

The sample IDs displayed in **S2 Data** are not known outside the research groups and cannot be used to reidentify any subject.

### Analysis of public SARS-CoV-2 genomes

All available SARS-CoV-2 data (402,899 genomes) were downloaded on January 22, 2021 from GISAID and evaluated for the presence of ORF1a Δ3675–3677 and spike Δ69–70. Phylogenetic analysis of a subset of 4,046 SARS-CoV-2 genomes was performed using Nextstrain [14], downsampled as shown using the "global build" on January 22, 2021 (https://nextstrain.org/ncov/global). A list of SARS-CoV-2 genomes used in the analysis is available in **S1 Data**.

### Multiplex RT-qPCR with probes

A detailed protocol of our multiplexed RT-qPCR to screen for SARS-COV-2 B.1.1.7, B.1.351, and P.1 VOC can be found on protocols.io [17]. In brief, our multiplex RT-qPCR assay consists of the CDC N1 [15] and the newly designed ORF1a Δ3675–3677 and spike Δ69–70 primer–probe sets (**S2 Table**). We used the NEB Luna universal probe 1-Step RT-qPCR kit

with 200 nM of N1 primers, 100 nM of N1 probe, 400 nM of the ORF1a and spike primers, 200 nM of ORF1a and spike probes, and 5 μL of nucleic acid in a total reaction volume of 20 μL. Thermocycler conditions were reverse transcription for 10 minutes at 55˚C, initial denaturation for 1 minute at 95˚C, followed by 40 cycles of 10 seconds at 95˚C and 30 seconds at 55˚C. During initial validation, we ran the PCR for 45 cycles. Differentiation between VOC is based on target failure of the ORF1a and/or spike primer–probe sets (S3 Table).

## Limit of detection

We used Twist synthetic SARS-CoV-2 RNA controls 2 (GenBank ID: MN908947.3; GISAID ID: Wuhan-Hu-1) and control 14 (GenBank ID: EPI_ISL_710528; GISAID ID: England/205041766/2020) to determine the limit of detection of the screening RT-qPCR assay. We tested a 2-fold dilution series from 100 copies/μL to 1 copy/μL for both RNA controls in triplicate and confirmed that the lowest concentration that was detected in all 3 replicates by 20 additional replicates.

## Validation and sequence confirmation

**United States.** We validated our approach using known SARS-CoV-2 positive clinical samples. Briefly, we extracted nucleic acid from 300 μL viral transport medium from nasopharyngeal swabs and eluted in 75 μL using the MagMAX viral/pathogen nucleic acid isolation kit (Thermo Fisher Scientific, Waltham, MA, United States). Extracted nucleic acid was tested by our multiplexed RT-qPCR assay and then sequenced using the Illumina COVIDSeq Test RUO version for the NovaSeq (paired-end 150), or using a slightly modified ARTIC Network nCoV-2019 sequencing protocol for the Oxford Nanopore MinION [20,21]. These modifications include extending incubation periods of ligation reactions and including a bead-based clean-up step following dA-tailing. MinION sequencing runs were monitored using RAMPART [22]. Consensus sequences were generated using the ARTIC Network bioinformatics pipeline and lineages were assigned using Pangolin v.2.0 [23,24]. GISAID accession numbers for all SARS-CoV-2 genomes used to validate our approach are listed in S2 Data.

**Brazil.** To validate the detection of the P.1 lineage, we selected 23 samples (16 P.1 and 7 others) that had been sequenced using the ARTIC protocol on the MinION sequencing platform, as previously described [12,25]. In brief, viral RNA was isolated from RT-PCR positive samples using QIAamp Viral RNA Mini kit (Qiagen, Hilden, Germany), following the manufacturer's instructions. cDNA was synthesized with random hexamers and the Protoscript II First Strand cDNA synthesis Kit (New England Biolabs, Ipswich, MA, United States). Whole genome multiplex-PCR amplification was then conducted using the ARTIC network SARS-CoV-2 V3 primer scheme with the Q5 High-Fidelity DNA polymerase (New England Biolabs). Multiplex-PCR products were purified by using AmpureXP beads (Beckman Coulter, Brea, CA, United States), and quantification was carried out using the Qubit dsDNA High Sensitivity assay on the Qubit 3.0 (Life Technologies, Carlsbad, CA, United States). Samples were then normalized in an equimolar proportion of 10 ng per sample. After end repair and dA tailing, DNA fragments were barcoded using the EXP-NBD104 (1–12) and EXP-NBD114 (13–24) Native Barcoding Kits (Oxford Nanopore Technologies, Oxford, United Kingdom). Barcoded samples were pooled together and sequencing adapter ligation was performed using the SQK-LSK 109 Kit (Oxford Nanopore Technologies). Sequencing libraries were loaded onto an R9.4.1 flow-cell (Oxford NanoporeTechnologies) and sequenced using MinKNOW version 20.10.3 (Oxford Nanopore Technologies). We tested RNA from sequenced samples with the multiplex RT-qPCR as described above using the Applied Biosystems 7500 real-time PCR machine (Thermo Fisher Scientific) and a lowered threshold of Ct 30.

**South Africa.** We extracted nucleic acid using the Chemagic 360 (PerkinElmer, Waltham, MA, United States). Briefly, 200 μl of viral transport medium from each swab sample was extracted and eluted in 100 μl using the Viral NA/gDNA kit. Complementary DNA (cDNA) synthesis, PCR, whole genome sequencing, and genome assembly was done as previously described in detail using the ARCTIC protocol [11,20]. Out of the sequenced samples, we selected 24 B.1.351 samples and 24 samples belonging to other lineages to validate the RT-qPCR assay. We adapted the protocol by using the TaqPath 1-Step multiplex master mix (Thermo Fisher Scientific) with 5 μl of extracted nucleic acid in a total reaction volume of 20 μl. Samples were amplified using the QuantStudio 7 Flex Real-Time PCR System using the following PCR conditions: Uracil-N-glycosylase (UNG) incubation for 2 minutes at 25˚C, reverse transcription for 15 minutes at 50˚C, polymerase activation for 2 minutes at 95˚C, followed by 40 cycles of amplification at 95˚C for 3 seconds and 55˚C for 30 seconds.

## Supporting information

**S1 Table. Summary of SARS-CoV-2 genomes with the ORF1a 3675–3677 and/or Spike 69–70 deletions (GISAID on January 21, 2020).** SARS-CoV-2, Severe Acute Respiratory Syndrome Coronavirus 2.
(DOCX)

**S2 Table. Primers and probes used in the multiplexed RT-qPCR variant screening assay.**
RT-qPCR, reverse transcription quantitative PCR.
(DOCX)

**S3 Table. Interpretation of results from the multiplexed RT-qPCR variant screening assay.**
RT-qPCR, reverse transcription quantitative PCR.
(DOCX)

**S1 Data. Underlying data for Fig 1.**
(XLSX)

**S2 Data. Underlying data for Fig 2.**
(XLSX)

## Acknowledgments

We thank A. Brito, C. Chiu, A. Lauring, A. Altajar, D. Comstock, P. Jack, S. Taylor, and V. Parsons for diagnostic samples, clinical support, and/or discussion. A list of acknowledgements for the SARS-CoV-2 data used in Fig 1 can be found in the S1 Data.

Members of Brazil-UK CADDE Genomic Network: Flavia Cristina da Silva Sales, Mariana Severo Ramundo, Darlan S. Candido, Camila Alves Maia Silva, Mariana Cardoso de Pinho, Thais de Moura Coletti, Pâmela dos Santos Andrade, Leandro Menezes de Souza, Esmênia Coelho Rocha, Ana Carolina Gomes Jardim, Erika Manuli, Nelson Gaburo Jr, Celso Granato, José Eduardo Levi, Silvia Costa, William Marciel de Souza, Maria Anice Salum, Rafael Pereira, Andreza de Souza, Lucy E. Matkin, Mauricio L. Nogueria, Anna Sara Levin, Philippe Mayaud, Neal Alexander, Renato Souza, Andre Luis Acosta, Carlos Prete, Joshua Quick, Oliver Brady, Janey Messina, Moritz Kraemer, Nelson da Cruz Gouveia, Izabel Oliva, Marcilio de Souza, Carolina Lazari, Cecila Salete Alencar, Julien Thézé, Lewis Buss, Leonardo Araujo, Mariana S. Cunha, Nicholas J Loman, Oliver G. Pybus, and Renato S. Aguiar.

Members of Network for Genomic Surveillance in South Africa (NGS-SA): Eduan Wilkinson, Nokukhanya Msomi, Arash Iranzadeh, Vagner Fonseca, Deelan Doolabh, Emmanuel James San, Koleka Mlisana, Anne von Gottberg, Sibongile Walaza, Mushal Allam, Arshad

Ismail, Thabo Mohale, Allison J. Glass, Susan Engelbrecht, Gert Van Zyl, Wolfgang Preiser, Francesco Petruccione, Alex Sigal, Diana Hardie, Gert Marais, Marvin Hsiao, Stephen Korsman, Mary-Ann Davies, Lynn Tyers, Innocent Mudau, Denis York, Caroline Maslo, Dominique Goedhals, Shareef Abrahams, Oluwakemi Laguda-Akingba, Arghavan Alisoltani-Dehkordi, Adam Godzik, Constantinos Kurt Wibmer, Bryan Trevor Sewell, José Lourenço, Sergei L. Kosakovsky Pond, Steven Weaver, Marta Giovanetti, Luiz Carlos Junior Alcantara, Darren Martin, Jinal N. Bhiman, and Carolyn Williamson.

## Author Contributions

**Conceptualization:** Chantal B. F. Vogels, Richard A. Neher, Joseph R. Fauver, Nathan D. Grubaugh.

**Data curation:** Chantal B. F. Vogels.

**Formal analysis:** Chantal B. F. Vogels, Mallery I. Breban, Isabel M. Ott, Tara Alpert, Mary E. Petrone, Anne E. Watkins, Chaney C. Kalinich, Rebecca Earnest, Jessica E. Rothman, Jaqueline Goes de Jesus, Ingra Morales Claro, Giulia Magalhães Ferreira, Myuki A. E. Crispim, Lavanya Singh, Houriiyah Tegally, Ugochukwu J. Anyaneji, Tulio de Oliveira, Nuno R. Faria, Ester C. Sabino, Richard A. Neher, Joseph R. Fauver, Nathan D. Grubaugh.

**Funding acquisition:** Nathan D. Grubaugh.

**Investigation:** Chantal B. F. Vogels, Mallery I. Breban, Isabel M. Ott, Tara Alpert, Mary E. Petrone, Anne E. Watkins, Chaney C. Kalinich, Rebecca Earnest, Jessica E. Rothman, Jaqueline Goes de Jesus, Ingra Morales Claro, Giulia Magalhães Ferreira, Myuki A. E. Crispim, Lavanya Singh, Houriiyah Tegally, Ugochukwu J. Anyaneji, Emma B. Hodcroft, Christopher E. Mason, Gaurav Khullar, Jessica Metti, Joel T. Dudley, Matthew J. MacKay, Megan Nash, Jianhui Wang, Chen Liu, Pei Hui, Steven Murphy, Caleb Neal, Eva Laszlo, Marie L. Landry, Anthony Muyombwe, Randy Downing, Jafar Razeq, Tulio de Oliveira, Nuno R. Faria, Ester C. Sabino, Richard A. Neher, Joseph R. Fauver, Nathan D. Grubaugh.

**Methodology:** Chantal B. F. Vogels, Joseph R. Fauver, Nathan D. Grubaugh.

**Resources:** Jaqueline Goes de Jesus, Ingra Morales Claro, Giulia Magalhães Ferreira, Myuki A. E. Crispim, Lavanya Singh, Houriiyah Tegally, Ugochukwu J. Anyaneji, Christopher E. Mason, Gaurav Khullar, Jessica Metti, Joel T. Dudley, Matthew J. MacKay, Megan Nash, Jianhui Wang, Chen Liu, Pei Hui, Steven Murphy, Caleb Neal, Eva Laszlo, Marie L. Landry, Anthony Muyombwe, Randy Downing, Jafar Razeq, Tulio de Oliveira, Nuno R. Faria, Ester C. Sabino, Nathan D. Grubaugh.

**Supervision:** Joseph R. Fauver, Nathan D. Grubaugh.

**Validation:** Chantal B. F. Vogels, Mallery I. Breban, Jaqueline Goes de Jesus, Ingra Morales Claro, Giulia Magalhães Ferreira, Myuki A. E. Crispim, Lavanya Singh, Houriiyah Tegally, Ugochukwu J. Anyaneji, Tulio de Oliveira, Nuno R. Faria, Ester C. Sabino, Joseph R. Fauver, Nathan D. Grubaugh.

**Visualization:** Chantal B. F. Vogels, Emma B. Hodcroft, Nathan D. Grubaugh.

**Writing – original draft:** Chantal B. F. Vogels, Nathan D. Grubaugh.

**Writing – review & editing:** Chantal B. F. Vogels, Mallery I. Breban, Isabel M. Ott, Tara Alpert, Mary E. Petrone, Anne E. Watkins, Chaney C. Kalinich, Rebecca Earnest, Jessica E. Rothman, Jaqueline Goes de Jesus, Ingra Morales Claro, Giulia Magalhães Ferreira, Myuki A. E. Crispim, Lavanya Singh, Houriiyah Tegally, Ugochukwu J. Anyaneji, Emma B.

Hodcroft, Christopher E. Mason, Gaurav Khullar, Jessica Metti, Joel T. Dudley, Matthew J. MacKay, Megan Nash, Jianhui Wang, Chen Liu, Pei Hui, Steven Murphy, Caleb Neal, Eva Laszlo, Marie L. Landry, Anthony Muyombwe, Randy Downing, Jafar Razeq, Tulio de Oliveira, Nuno R. Faria, Ester C. Sabino, Richard A. Neher, Joseph R. Fauver, Nathan D. Grubaugh.

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
