## [Editor Report · Decision Letter 0]

12 Mar 2021

Dear Dr. Vogels, 

Thank you for submitting your manuscript entitled "PCR assay to enhance global surveillance for SARS-CoV-2 variants of concern" for consideration as a Short Reports by PLOS Biology.

Your manuscript has now been evaluated by the PLOS Biology editorial staff and I am writing to let you know that we would like to send your submission out for external peer review.

Please re-submit your manuscript within two working days, i.e. by Mar 14 2021 11:59PM.

Kind regards,

Paula

---

Associate Editor

PLOS Biology

---

## [Decision Letter · Decision Letter 1]

9 Apr 2021

Dear Dr. Vogels,

Thank you very much for submitting your manuscript "PCR assay to enhance global surveillance for SARS-CoV-2 variants of concern" for consideration as a Research Article by PLOS Biology. As with all papers reviewed by the journal, yours was evaluated by the PLOS Biology editors as well as by an Academic Editor with relevant expertise and by independent reviewers. The reviewers appreciated the attention to an important topic. 

Based on the reviews, we will probably accept this manuscript for publication, provided you satisfactorily address any points raised by the reviewers. 

We suggest a change of title in order to make it a bit more declarative, but please feel free to modify it: "Multiplex qPCR discriminates variants of concern to enhance global surveillance of SARS-CoV-2"

We expect to receive your revised manuscript within two weeks. 

*Published Peer Review History*

*Early Version*

Sincerely,

Paula

---

Associate Editor,

pjaureguionieva@plos.org,

PLOS Biology

Reviewer remarks:

Reviewer #1: Coronavirus with interest in diagnosis/detection.

Reviewer #2: Emerging viruses.

Reviewer #1: "PCR assay to enhance global surveillance for SARS-CoV-2 variants of concern," by Vogels et al describes development and testing of a multiplex qPCR screening approach to identify probable SARS2 variant of concern samples. While the late 2020 discovery that S gene dropout PCR tests could indicate B117 variant infection was fortuitous, those results were far from definitive and not capable of identifying all variants of concern. By targeting an Orf1a deletion in addition to the spike deletion, a more targeted and broadly useful assay can be performed while using the same sample types and overall technology. The benefits and limitations of this assay are clearly described and the sample testing was robust. As a new reviewer of this revised manuscript I am not able to see the previous reviews to assess how they were addressed; however; I have no concerns with this work and encourage speedy publication.

Reviewer #2: Tracking the emergence and spread of SARS-CoV2 variants has great public health importance, and the paper by Vogels and colleagues describes a real-time PCR-based approach to identify key variants of interest/concern; particularly B.1.1.7, B.135 and P.1. The authors state that such an approach would be of significant benefit where access to NGS analysis was limited, and even in resource-rich regions, where NGS was readily available, such an approach could be used to effectively triage samples for sequencing.

The approach take was robust and the validation, albeit on a relatively small number of clincial samples (especially samples from Brazil and South Africa), convincing. Whilst the methods were able to identify target variants, the discriminatory power was less than 90% for some variants, meaning that downstream sequence validation would be required. Similarly, the authors also state that the multi-plex PCR would not be able to replace existing diagnostic approaches, due to non-specific auto-fluourescence. Therefore, this method would be useful as an additional step between clinical diagnosis and downstream sequencing but would not be able to replace either.

It was unclear to me why this paper would be suitable for publication in PLoS Biology - it would be much more suited to a journal specialising in viral diagnostics.

---

## [Editor Report · Decision Letter 2]

16 Apr 2021

Dear Dr. Vogels,

On behalf of my colleagues and the Academic Editor, Bill Sugden, I am pleased to say that we can in principle offer to publish your Methods and Resources paper "Multiplex qPCR discriminates variants of concern to enhance global surveillance of SARS-CoV-2" in PLOS Biology, provided you address any remaining formatting and reporting issues. These will be detailed in an email that will follow this letter and that you will usually receive within 2-3 business days, during which time no action is required from you. Please note that we will not be able to formally accept your manuscript and schedule it for publication until you have made the required changes.

PRESS

Thank you again for supporting Open Access publishing. We look forward to publishing your paper in PLOS Biology. 

Sincerely, 

Paula 

---

Paula Jauregui, PhD 

Associate Editor 

PLOS Biology